# Effects of Bonding Materials on Optical–Thermal Performances and High-Temperature Reliability of High-Power LED

**DOI:** 10.3390/mi13060958

**Published:** 2022-06-17

**Authors:** Jiaxin Liu, Yun Mou, Yueming Huang, Jiuzhou Zhao, Yang Peng, Mingxiang Chen

**Affiliations:** 1School of Mechanical Science and Engineering, Huazhong University of Science and Technology, Wuhan 430074, China; stu_liujx@163.com (J.L.); mouluck@163.com (Y.M.); 2School of Aerospace Engineering, Huazhong University of Science and Technology, Wuhan 430074, China; U201811242@hust.edu.cn (Y.H.); zhaojiuzhou2022@163.com (J.Z.)

**Keywords:** Ag nanoparticle, sintering, bonding material, high-temperature reliability, high-power LED

## Abstract

The die-bonding layer between chips and substrate determinates the heat conduction efficiency of high-power LED. Sn-based solder, AuSn20 eutectic, and nano-Ag paste were widely applied to LED interconnection. In this paper, the optical–thermal performances and high-temperature reliability of LED with these bonding materials have systematically compared and studied. The thermal conductivity, electrical resistivity, and mechanical property of these bonding materials were characterized. The LED module packaged with nano-Ag has a minimum working temperature of 21.5 °C. The total thermal resistance of LED packaged with nano-Ag, Au80Sn20, and SAC305 is 4.82, 7.84, and 8.75 K/W, respectively, which is 4.72, 6.14, and 7.84 K/W higher after aging for 500 h. Meanwhile, the junction temperature change of these LEDs increases from 2.33, 3.76, and 4.25 °C to 4.34, 4.81, and 6.41 °C after aging, respectively. The thermal resistance of the nano-Ag, Au80Sn20 and SAC305 layer after aging is 1.5%, 65.7%, and 151.5% higher than before aging, respectively. After aging, the LED bonded with nano-Ag has the better optical performances in spectral intensity and light output power, which indicates its excellent heat dissipation can improve the light efficiency. These results demonstrate the nano-Ag bonding material could enhance the optical-thermal performances and high-temperature reliability of high-power LED.

## 1. Introduction

Light-emitting diode (LED) has been considered as a promising solid-state light source owing to its high light efficiency, low power consumption, better environmental protection, long lifetime, and small size [1,2,3]. With the increasing demand of high-brightness lighting in our daily life, such as outdoor lighting, landscape lighting, car lighting, and special lighting, the input current of LED chips is increased and the multi-chips packaging structure is used to product high-power LEDs [4,5]. As we know, the electro-optical conversion efficiency of an LED chip is no more than 60%, and the residual input electrical energy is converted to thermal energy in LED chip. The accumulated heat leads to the high junction temperature (>150 °C) for high-power LED, which not only deteriorates the luminescence property of LED chip but also generates the thermal degradation and failure of packaging materials [6,7]. Thus, the optical–thermal performances of a high-power LED seriously depend on its heat dissipation.

In the LED packaging structure, LED chips are bonded on a packaging substrate, and most of heat is propagated to the surrounding environment through the heat conduction characteristic of packaging substrate. The packaging substrate with high thermal conductivity (such as metal substrate and ceramic substrate) can enhance the heat dissipation and reduce the heat-induced performance degradation of high-power LED [8]. It should be noted that the die-bonding layer between LED chips and substrate determinates the heat conduction efficiency of high-power LED originating from its heat conduction path [9,10,11]. Currently, common die-bonding materials are Sn-Ag-Cu solder (SAC305) and Au80Sn20 eutectic solder (AuSn), which realize reliable chip fixation under various bonding temperatures and pressures [12,13,14]. Although the SAC305 has the advantages of good solderability, low melting point, and low cost, it struggles to withstand high operation temperature and the die-bonding layer has poor electrical conductivity and large brittleness [15,16]. The AuSn enhances the tolerable temperature and bonding strength of die-bonding layer, but it requires high-cost equipment and the high process temperature easy to damage chips [17]. Recently, a new die-bonding material, nano-silver paste (nano-Ag), has been developed owing to the small size effect of nanomaterials [18,19,20]. Although the nano-Ag achieves a low bonding temperature, high working temperature, and high electrical/thermal conductivity, the long-term reliability of the bonding layer should be considered owing to the thermoelectric effect of Ag ions [21,22]. Furthermore, the optical–thermal performances of high-power LEDs with these bonding materials and their reliability have not systematically compared and studied yet. Therefore, it is necessary to investigate the effects of different bonding materials on the optical–thermal performances and high-temperature reliability of high-power LED.

In this work, we investigated the effects of bonding materials on the optical–thermal performances and high-temperature reliability of high-power LEDs. Performance of LEDs with Sn-based solder, AuSn20 eutectic, and nano-Ag paste were systematically studied. The thermal conductivity, electrical resistivity, and mechanical properties of these bonding materials were characterized. The thermal resistance, junction temperature change, spectral intensity, and light output power before and after aging were measured and compared.

## 2. Materials and Methods

In this experiment, the bonding materials we used were commercial products. The nano-Ag paste was purchased from Advanced Connection Technology Co., Ltd. (Shenzhen, China). The Au80Sn20 soldering flakes were purchased from Turing Electronic Technology Co., Ltd. (Shanxi, China). The chip used was a high-power vertical packaging blue LED chip (1 W) with AlN DPC ceramic substrate. The thermoelectric separation heat sink used was hexagonal Cu substrate. The dehydrated ethanol and acetone were purchased from Sinopharm Chemical Reagent Co., Ltd. (China).

Figure 1 presents the preparation process of a high-power LED with nano-Ag paste. Firstly, the ceramic and Cu substrates were ultrasonically cleaned with acetone and anhydrous ethanol to remove the impurities on the surface of metal pad. Then, nano-Ag paste was coated on the surface pad of the Cu substrate by screen printing with a thickness of 100 μm. Subsequently, the metal pad on the back of the DPC ceramic substrate was aligned with the Ag layer and mounted on the Cu substrate. The sintering process was completed in an oven. After holding at 120 °C for 10 min, the LED sample was sintered at 220 °C for 60 min. Finally, the positive and negative electrodes of the Cu substrate were welded to facilitate following thermal and optical performance testing of LED. The packaging process of Au80Sn20 and SAC305 was the same as that of nano-Ag paste. AuSn was welded in the vacuum furnace, and SAC was interconnected in the traditional reflow oven. In order to test the electrical resistivity of nano-Ag paste after sintering, nano-Ag paste was coated on the quartz glass and sintered at 220 °C. To compare the mechanical property of the bonding materials, bonding samples were fabricated with nano-Ag paste, Au80Sn20 and SAC305. The upper and bottom bonded Cu substrates (with 2 μm Ag layer) used in our experiment were 3 × 3 × 2 mm^3^ and 8 × 8 × 2 mm^3^ in size, respectively.

The film thickness of sintered nano-Ag was measured by a surface profiler (ET4000 Series), and the square resistance of sintered Ag film was tested by four-probe tester (Probes RST-8, Shenzhen, China). The shear strengths of bonded joints were tested by Multifunctional Shear Force Tester (Dage 4000Plus Bond, UK). The shear speed was set as 0.2 mm/min. A scanning electron microscope (SEM, FEI Nova NanoSEM 450, USA) was used to observe the cross-sectional microstructure and fracture surface of the bonding joints. The cross-sectional structure of the high-power LED was observed by optical microscope (KEYENCE, VHX-1000, Japan). Thermal imaging system (FLIR, E63, USA) was used to record the surface operating temperature of the high-power LED. The junction temperature change and structural thermal resistance of LED devices with different bonding materials were tested by the thermal resistance tester (T3STER-Master, Mentor Graphics, Germany), the current was set as 1 mA. Photoelectric analysis system (HAAS-2000, EVERFINE, China) was used to test the emission spectrum and light output power of LED samples.

## 3. Results

### 3.1. Properties of Different Bonding Materials

Figure 2a shows the thermal conductivity and electrical resistivity of different bonding materials. The thermal conductivity of nano-Ag, Au80Sn20, and SAC305 is 140, 57, and 33 W/(m·k), respectively. The electrical resistivity of nano-Ag sintered at 220 °C is 4.9 μΩ·cm, and that of the Au80Sn20 and SAC305 is 22.4 and 10.8 μΩ·cm, respectively. After sintering at 220 °C, the electrical resistivity of nano-Ag is close to the theoretical value of metal Ag, indicating a complete sintering. It can be seen that the nano-Ag paste has the best thermal conductivity and lowest electrical resistivity. Figure 2b presents the shear strengths of the bonding joints with different bonding materials. The shear strength of the joints with nano-Ag, Au80Sn20, and SAC305 is 20.3, 33.3, and 17.4 MPa, respectively. Au80Sn20 interconnection layer has the highest shear strength, which may due to the inherent low porosity and high density. The higher temperature will bring higher sintering driving force and accelerate the sintering and diffusion of nanoparticles. Therefore, it could be that when the sintering temperature continues to rise, the shear strength of the nano-Ag joint would continue to increase.

Figure 3a,d show the cross-sectional and fracture surface morphologies of the bonding joint with sintered nano-Ag. The Ag layer was well bonded with the Cu substrates; no cracks and defects could be found at the bonding interface. There were a few pores in the interconnection layer, which was due to the volatilization of organic solvent during sintering. The ductile fracture morphology of Ag nanoparticles could be clearly seen from the fracture surface. The particles were melted and sintered at high temperature to form sintering neck, and then elongated under the shear force until ductile fracture. Figure 3b,e and Figure 3c,f are the cross-sectional and fracture surface morphologies of the joints with Au80Sn20, and SAC305, respectively. The interconnection bondline and substrate formed nice metallurgical bonding without cracks and gaps. Mixed fracture occurred in the Au80Sn20 joint, while the ductile fracture was located between Au_5_Sn phase and Cu substrate, and the brittle fracture was located between Au_5_Sn phases. The grains became coarse and transformed into IMC brittle phase under the high temperature, forming thermal stress concentration area, which would become the crack source. The crack propagated to the Cu substrate with lower shear modulus and produced ductile fracture. The bonding joint with SAC305 also presents a ductile fracture morphology. However, a number of holes were left in the bondline caused by the volatilization of solvent and flux in SAC solder.

### 3.2. Characteristics of LED with Different Bonding Materials

Figure 4a presents the cross-sectional image of the LED module packaging with nano-Ag. It is clear that the high-power LED chip is a vertical packaging structure. The LED module is composed of chip, metal layer, AlN ceramic substrate, metal pad, nano-Ag bonding layer, and Cu substrate (from top to bottom). Figure 4b shows the magnified view of the red-box region, which is the microstructure of the nano-Ag bonding layer. The bonding layer has a dense structure without internal holes and cracks. Meanwhile, the interfaces between the nano-Ag layer and the Cu substrates are clear with a good metallurgical bonding. There are no defects at the interface, which indicating that the LED module sintered by nano-Ag paste has a good packaging quality.

The working temperatures of LED module packaged with different bonding materials were tested. The LED device was mounted on the heat sink by thermal grease. The ambient temperature was controlled at 25 °C, and the height of the thermal imaging system was 0.5 m from the samples. Figure 5 shows the thermal images of LED under the driving current of 350 mA after 5 min of operation. The working temperature of the nano-Ag sample is 21.5 °C, which is 16.0% lower than that of Au80Sn20 sample (25.6 °C) and 23.8% lower than that of SAC305 sample (28.2 °C). Under the same conditions, the LED packaged with nano-Ag paste has the lowest operating temperature, indicating a best heat dissipation.

### 3.3. High-Temperature Reliability of High-Power LED

In order to investigate the effect of different bonding materials on the reliability of high-power LED, the three LED samples were placed in an oven for a high-temperature aging test. The aging temperature and driving current was 100 °C and 350 mA, respectively. The optical and thermal properties of the LED samples were tested at 100, 200, 300, 400, and 500 h during aging. Figure 6 shows the differential thermal resistance of LED samples with different bonding materials after aging. The total thermal resistance of LED device was minimum before aging, and then increased with the increasing aging time. The thermal resistance of the LED packaging with nano-Ag paste varied little, while the LED samples packaging with Au20Sn80 and SAC305 varied greatly after 500 h, which indicates that the nano-Ag has a good high-temperature stability.

Figure 7a shows the differential thermal resistance of LED with different bonding materials before aging. The total thermal resistance of LED packaged with nano-Ag, Au80Sn20, and SAC305 is 4.82, 7.84, and 8.75 K/W, respectively. The total thermal resistance of nano-Ag sample is 38.5% and 44.9% lower than that of the Au80Sn20 and SAC305 sample. After aging for 500 h, the total thermal resistance of nano-Ag, Au80Sn20, and SAC305 sample is 9.54, 13.98, and 16.59 K/W, which are 4.72, 6.14, and 7.84 K/W higher than before, as shown in Figure 7c. The T3ster calculated the corresponding relationship between temperature change over time through the temperature coefficient (mV/°C), and its junction temperature change curves are shown in Figure 7b. The junction temperature change of the LED packaged with nano-Ag, Au80Sn20, and SAC305 was 2.33, 3.76, and 4.25 °C before aging, which increased to 4.34, 4.81, and 6.41 °C after aging, respectively. Through the aging test, the thermal resistance and junction temperature of the devices all increased, indicating that the long-term high-temperature working could accelerate the aging of packaging materials and reduce the heat dissipation efficiency of device. In this process, the LED packaging with nano-Ag paste always maintained the lowest total thermal resistance and junction temperature, demonstrating its excellent thermal conductivity and high temperature stability.

The region between each peak of wave in the differential thermal resistance curve represents the thermal resistance value of the corresponding LED structure. The peaks of wave divide the curve from left to right into five regions, corresponding to the resistance value of LED chip, silver paste, AlN ceramic substrate, bonding layer and hexagonal Cu substrate, respectively. Figure 8a presents the thermal resistance of bonding layer under different aging time. As the aging time increased, the thermal resistance of nano-Ag layer was 0.67, 0.43, 0.56, 0.58, 0.65, and 0.6, respectively. The thermal resistance decreased first and then increased, which was maybe caused by the deep sintering of Ag nanoparticles at high temperature. The particles continued to grow and form sintering neck, leading to a complete sintering and thermal conductivity improvement. With further aging, the porosity and thermal resistance of Ag nanoparticles increased gradually. The thermal resistance of the Au80Sn20 and SAC305 bonding layer increased with the increasing aging time. Meanwhile, the thermal resistance of SAC305 layer was always higher than that of the Au80Sn20 layer. During the long-term high-temperature operation, diffusion and migration occurred in the SAC305 bondline, resulting the Kirkendall voids, which enhances the interface thermal resistance. The thermal resistance of the nano-Ag, Au80Sn20 and SAC305 layer after aging was 1.5%, 65.7%, and 151.5% higher than before aging, respectively. Figure 8b shows the junction temperature change of LED under different aging time. The junction temperature increased with the increasing aging time. The LED packaging with nano-Ag possesses the lowest junction temperature, which has a better thermal conductivity.

Figure 9a shows the spectra of LED with nano-Ag under different aging time. The driving current was 350 mA. As the aging time increased to 500 h, the spectral intensity increased first and then decreased, which is consistent with the rule above. Figure 9b,c display the spectra of Au80Sn20 and SAC305 sample, which spectral intensity decreased gradually with the increasing aging time. The spectral intensity of nano-Ag sintered LED was always higher than those of the Au80Sn20 and SAC305 sample. This may be due to the good thermal conductivity of nano-Ag, which converts the total power more into the optical power, improving the spectral intensity and optical efficiency of LED.

Figure 10a shows the light-output power of LEDs under different aging times. The LED samples with different bonding materials had a similar light output power before aging. With the increase of aging time, the light-output power of LEDs with Au80Sn20 and SAC305 decreased gradually, while the light power of nano-Ag sample increased first and then decreased slowly. Figure 10b presents the light power attenuation after different aging times. The light power of the nano-Ag, Au80Sn20, and SAC305 sample after aging is 6.1%, 9.3%, and 9.7% lower than before aging, respectively. Nano-Ag has better heat dissipation performance and high-temperature resistant. The heat generated by LED in the lighting process could be better dissipated through the nano-Ag interconnection layer and improve the light efficiency.

### 3.4. Thermal Simulation of High-Power LED

In order to verify the effect of bonding material on the thermal resistance of LEDs, COMSOL Multiphysics software was used to analyze the temperature field distribution of LED model. To shorten the calculation time and reduce the amount of computation, the LED model was simplified. The effects of epitaxial materials (GaN), electrode materials and metal pad of substrates on heat transfer were ignored, and a half model was adopted for simulation calculation. Therefore, the LED model mainly includes LED chip, silver paste, AlN ceramic substrate, bonding layer, thermoelectric separation insulation layer, thermoelectric separation Cu substrates and air interface. The thickness and thermal conductivity parameters of packaging materials are shown in Table 1.

The simulated ambient temperature was set as 25 °C. The bottom surface of the thermoelectric separation Cu substrate was assumed to be forced convection heat transfer. The other contact boundaries with air were set as natural convection heat transfer, and the heat transfer coefficient was set as 2 W/(m^2^·K). The input power and optical power of LED sample were measured by integrating sphere under current of 1200 mA, and the difference was used as the simulated thermal power for simulation analysis. Figure 11 shows the temperature field distribution of LED packaging with different bonding materials. It is clear that the LED chips have the highest temperature. When the bonding materials are nano-Ag, Au80Sn20, and SAC305, the maximum temperature of the LED chip is 38.2 °C, 41.4 °C and 42.7 °C, respectively. Compared with Au80Sn20 and SAC305, nano-Ag paste is more conducive to the heat dissipation of LED chip, which is consistent with the experimental results.

## 4. Conclusions

In this paper, the optical–thermal performance and high-temperature reliability of high-power LED packaged with SAC305, Au80Sn20, and nano-Ag paste were systematically investigated. The nano-Ag paste has the highest thermal conductivity of 140 W/(m·k) and the lowest electrical resistivity of 4.9 μΩ·cm. The total thermal resistance and junction temperature change of LED increased with the increasing aging time. After aging at 100 °C for 500 h, the total thermal resistance of LED packaged with nano-Ag, Au80Sn20, and SAC305 increased from 4.82, 7.84, and 8.75 K/W to 9.54, 13.98, and 16.59 K/W, the junction temperature change increased from 2.33, 3.76, and 4.25 °C to 4.34, 4.81, and 6.41 °C, respectively. The thermal resistance of the bonding layer is 1.5%, 65.7%, and 151.5% higher than before aging, while the light-output power is 6.1%, 9.3%, and 9.7% lower than before aging, respectively. Nano-Ag has better heat dissipation ability and high-temperature resistance, which enhances the comprehensive performance of high-power LED.

## Figures and Tables

**Figure 1 micromachines-13-00958-f001:**
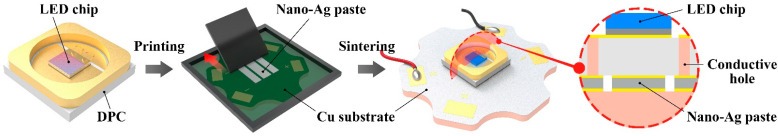
Preparation process of high-power LED with nano-Ag paste.

**Figure 2 micromachines-13-00958-f002:**
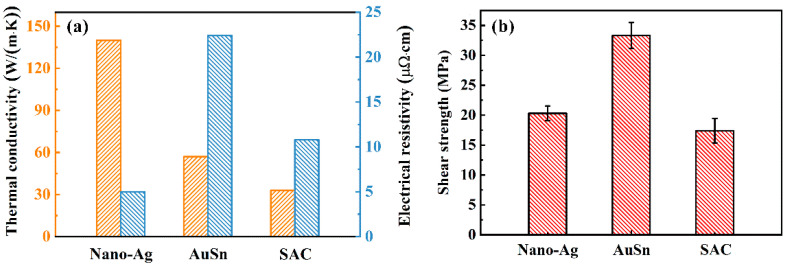
(**a**) Thermal conductivity and electrical resistivity of different bonding materials. (**b**) Shear strength of bonding joints bonded with different materials.

**Figure 3 micromachines-13-00958-f003:**
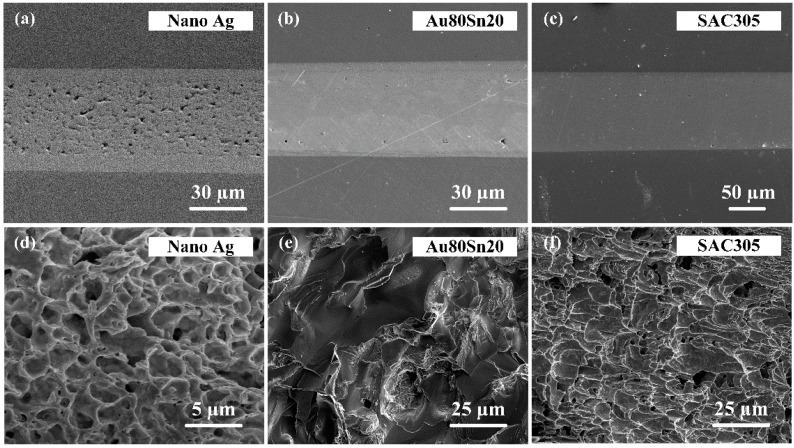
Cross-sectional of the bonding joints with (**a**) nano-Ag, (**b**) Au80Sn20, and (**c**) SAC305. (**d**–**f**) Fracture surface morphology of the bonding joints with different materials.

**Figure 4 micromachines-13-00958-f004:**
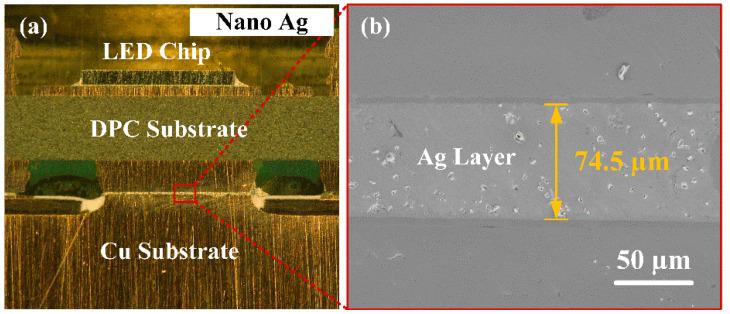
Cross-sectional images of (**a**) LED module and (**b**) nano-Ag bonding layer.

**Figure 5 micromachines-13-00958-f005:**
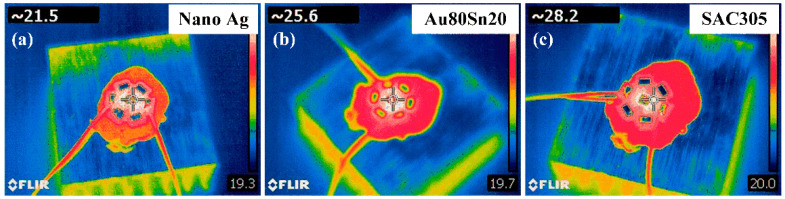
Thermal images of LED packaging with (**a**) nano-Ag, (**b**) Au80Sn20, and (**c**) SAC305.

**Figure 6 micromachines-13-00958-f006:**
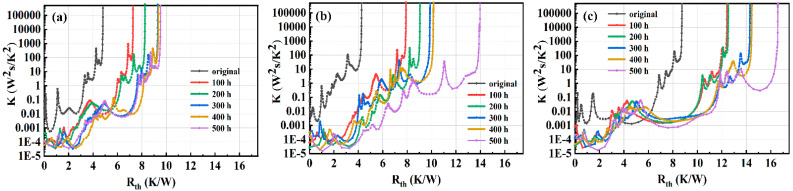
Differential thermal resistance of LED under different aging time packaging with (**a**) nano-Ag, (**b**) Au80Sn20, and (**c**) SAC305.

**Figure 7 micromachines-13-00958-f007:**
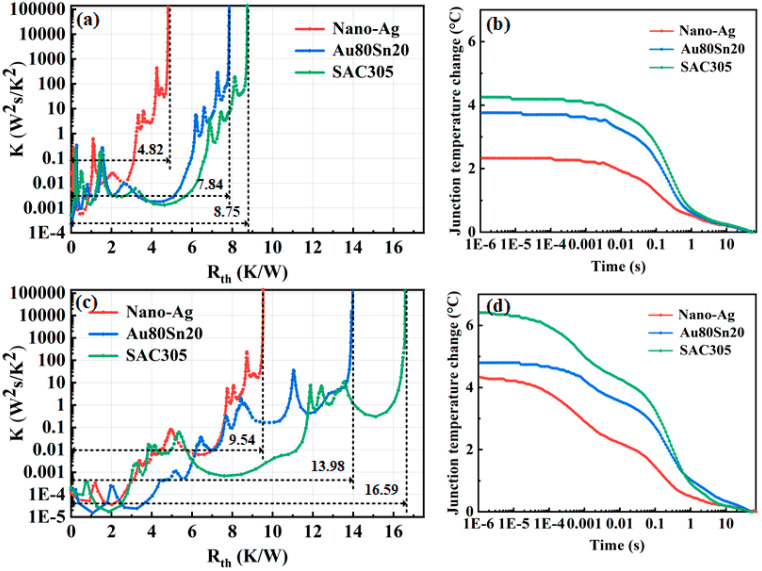
(**a**) Differential thermal resistance of LED with different bonding materials before aging. (**b**) Junction temperature change of LED with different bonding materials before aging. (**c**) Differential thermal resistance of LED with different bonding materials after aging. (**d**) Junction temperature change of LED with different bonding materials after aging.

**Figure 8 micromachines-13-00958-f008:**
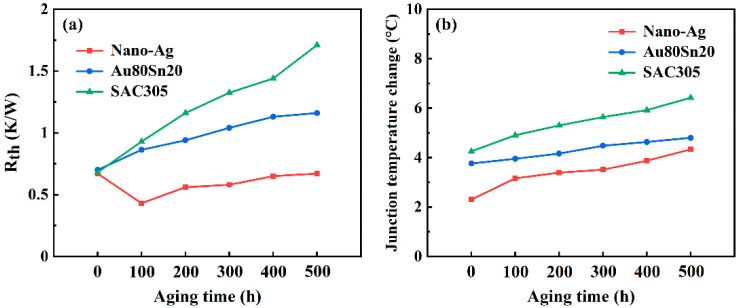
(**a**) Thermal resistance of bonding layer under different aging time. (**b**) Junction temperature change of LED under different aging times.

**Figure 9 micromachines-13-00958-f009:**
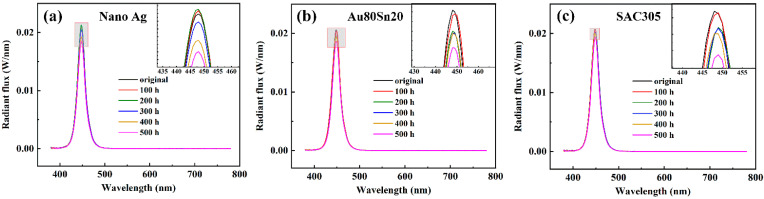
Spectra of LEDs under different aging time packaging with (**a**) nano-Ag, (**b**) Au80Sn20, and (**c**) SAC305.

**Figure 10 micromachines-13-00958-f010:**
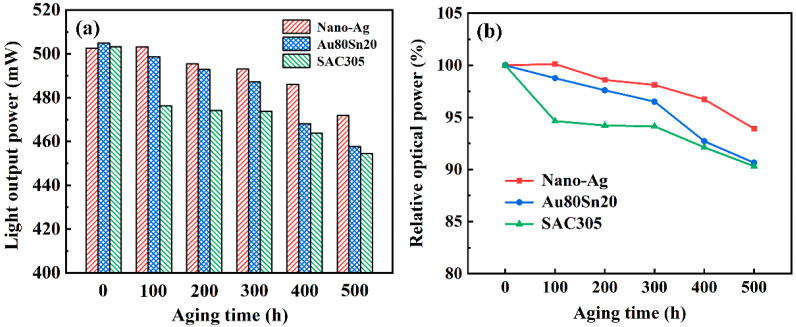
(**a**) Light output power and (**b**) relative optical power of LEDs under different aging times.

**Figure 11 micromachines-13-00958-f011:**
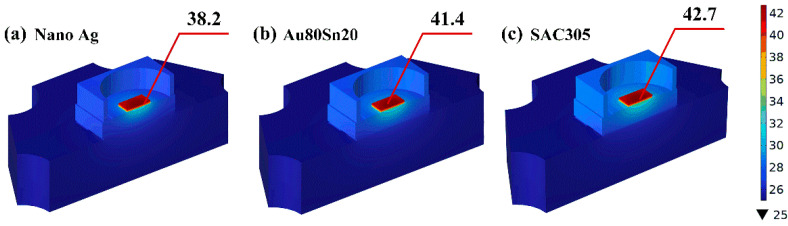
Temperature field distribution of LED packaging with different bonding materials.

**Table 1 micromachines-13-00958-t001:** Thickness and thermal conductivity parameters of packaging materials in simulation.

Packaging Materials	Thickness (μm)	Thermal Conductivity (W/(m·K))
LED chip	100	25
Cu dam	840	390
Silver paste	20	7.5
AlN ceramic substrate	635	170
Insulation layer	30	0.6
Cu substrate	1600	390
Nano-Ag paste	75	140
Au80Sn20	75	57
SAC305	75	33

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
