# Peer review of "Effects of Bonding Materials on Optical–Thermal Performances and High-Temperature Reliability of High-Power LED"

_micromachines, 2022, doi:10.3390/mi13060958_

Round 1

Reviewer 1 Report

Dear Autor, 

your manuscript deals with the comparison of three interconnection technologies according their ageing perfomance impacting LED system behaviour (thermal resistance and radiant flux). The choice of interconnection material can improve the thermal performance, but is not the weak point in an overall system view. Heat spreading by the used base plate or the used thermal interface material to the heat-sink has a higher impact. But nevertheless, the ageing behaviour of the investigated materials can affect the long therm reliability, but it is necessary to distinguish between the different failure mechanisme and the right ageing test. You have to investigate more in LED ageing, since there is a lot of existing literature.  To improve your investigations here are my questions and comments:  

1) line 37: high power LEDs have a max operating temperature of 150°C, junction temperature >300°C can not be possible

2) material & methods

+ which kind of high power LED was investigated? Since data sheet informations can improve your discussion.

+ the desciption of your used methods (Shear Test, thermal imaging, TI-Analysis, ...) must be improved by descibing the used boundary conditions and equations. Or use a reference when you are using some norms/standards.

+ you are using soldering and sintering. Cu-Cu bonding is a quite different interconnection technology

3) results

+ how the shear strength impacts the optical performance or temperature reliability? This comparison and the following fracture surface analysis makes no sense in the context of the further investigations.

+ thermal image analysis fits not with your investigated thermal conductivity analysis (ratio is not equal). How was your StarBoard mounted on the heatsink. The mounting and the used TIM (if you use one) has a higher impact and can lead to the differences. 

+ HTOL is not the right ageing method, to investigate in interconnection technologies, since the driving condition ages the LED device itself (see LM80 testing). And how the junction temperature looks like when your are using the same current, since the temperaure is the driving point for IMC growth and grain coarsening

+ you have to improve your T3ster measurements/analysis. (1) you have a Cth shift from initial to aged curves, (2) none aged parts like the LED module should always looks the same, (3) a comparison of the total Rth can not be downscaled to a pure interconnection ageing effect, (4) temperature transient (Figure 7d) showing double transient effect that means your HS cooling power is not enough, (5) your device reached not the steady state during down cooling, (6) ... --> i would strongly advise against interpreting these measurement series

+ the resolution of the T3ster system is not sufficient to analyse IMC growth and other texture changes. The changes in the microstructure as long as no crack or degradations occurs, will not really affect the Rth after 500h ageing

+ how the SF-calibration for the structure function analysis looks like?

+ based on your application/settings, the radiant flux interpretation is not plausible. The changes might come from the degradation of the LED-module itself (see LM80 testing)

Regards, the reviewer

Reviewer 2 Report

The manuscript makes comparison between three different materials for high power LED bonding, based on experimental work and simulation on real LED device with commercial software. The result leads to the conclusion that the nano-Ag is the most superior one among those three. The result of the manuscript is with practical use but somehow lack of scientific investigation and  deep understanding. Both academic contribution and technical breakthrough are limited, especially those three materials have been widely used in industry.  It is suggested that the mechanism which makes nano-Ag outperform over the others, even with aging, can be described and discussed deeply together with relevant material characterization, such as microscopic observation etc.

Round 2

Reviewer 1 Report

Dear Autor, 

you answered all the questions and have inserted the most necessary things according methods.  

Regards, the reviewer

Reviewer 2 Report

There is no much change and no new result added in the revised manuscript, but the authors' viewpoint on the contribution of the manuscript is accepted.